# Postpartum cytokine shifts and IL-10–mediated immune suppression in malaria-infected primigravid women

Ousmane Traore[1]*, Toussaint Rouamba[1], Serge Henri Zango[1], Hermann Sorgho[1], Innocent Valea[1], Maminata Traore-Coulibaly[1], Henk D. F. H. Schallig[2], Halidou Tinto[1]

**1** Institut de Recherche en Sciences de la Santé, Unité de Recherche Clinique de Nanoro (IRSS-URCN), Nanoro, Burkina Faso, **2** Department of Medical Microbiology and Infection Prevention, Laboratory for Experimental Parasitology, Amsterdam Institute for Infection and Immunity, Amsterdam University Medical Centres, Academic Medical Centre at the University of Amsterdam, Amsterdam, The Netherlands

* ousmane_tra@yahoo.fr

## Abstract

### Background

According to the World Health Organization's recent report, malaria remains a major health challenge during pregnancy and for postpartum women in endemic regions. While immune alterations during pregnancy are well characterized, postpartum cytokine dynamics and their impact on malaria susceptibility remain poorly defined. This study uniquely investigates how cytokine balance shifts, contribute to malaria susceptibility in primigravid women during the postpartum period.

### Methods

A total of 33 Burkinabè women were enrolled at delivery and followed up at 1 and 3 months postpartum. Plasma cytokine concentrations (IL-4, IL-6, IL-10, TNF-α, IFN-γ) were quantified by ELISA. Malaria infection was detected by PCR and microscopy. Statistical analyses included effect size calculations and cluster analyses to assess immune profiles.

### Results

At delivery, 48.5% of women tested positive for malaria by PCR. Malaria-infected women had significantly elevated IL-10 levels and a decreased IL-6:IL-10 ratio compared with non-infected women ($p = 0.005$). This anti-inflammatory shift persisted into the early postpartum period. Strong correlations were observed between IL-10 levels and malaria infection ($\sigma = 0.9$, $p < 0.001$). Of note, IL-4 also showed a significant effect, highlighting a complex immunoregulatory environment.

**Data availability statement:** All relevant data are within the paper and its Supporting Information file.

**Funding:** This study received financial support from the Belgium cooperation (DGD-ITM Framework Agreement 4 – FA4 2017-2021), and from the European Union (FP7-Health-F3-305662). The funders had no role in study design, data collection and analysis, decision to publish, or preparation of the manuscript.

**Competing interests:** The authors have declared that no competing interests exist.

## Conclusion

Our findings reveal, for the first time in a Sub-Saharan primigravid cohort, that an IL-10-dominant cytokine profile at delivery is strongly associated with postpartum malaria susceptibility. Modulating cytokine responses could represent a novel therapeutic approach to *improving* maternal health in malaria-endemic regions.

## Introduction

Malaria remains a major public health challenge, particularly among pregnant and postpartum women in endemic regions. Despite major advances in malaria control, maternal morbidity and mortality associated with *Plasmodium falciparum* infection persist, worsened by immune alterations during pregnancy and postpartum periods. Pregnancy induces a shift towards a Th2-dominant immune profile to protect the semi-allogeneic fetus [1–3]. However, this immunological modulation makes pregnant women more vulnerable to infections, particularly malaria. After delivery, the postpartum period is characterized by progressive immune reconstitution and rebalancing of Th1 and Th2 cytokine responses [4,5]. Recent studies suggest that the postpartum immune environment is distinct from that of pregnancy and non-pregnancy, but remains insufficiently understood, particularly with regard to malaria susceptibility [6–8]. Emerging data highlight the complex interplay between pro-inflammatory cytokines (e.g., IL-6, TNF-α, IFN-γ) and anti-inflammatory cytokines (e.g., IL-10, IL-4) in determining malaria outcomes [9,10]. Higher levels of IL-10 have been reported to be involved in both protection against excessive inflammation and facilitation of parasite persistence [11,12]. However, few studies have longitudinally assessed cytokine dynamics specifically during the postpartum window, particularly in primigravid women who are at highest risk of malaria and its complications [13,14]. Furthermore, while studies have explored placental malaria and perinatal transmission, there is a paucity of research on how postpartum cytokine balances might influence new malaria infections or recrudescence [15,16]. The need for finer characterization of immunoregulatory mechanisms during postpartum recovery in malaria-endemic settings is pinpointed by recent immunological investigations [17,18]. In the present study, we aim to address this critical gap by investigating the association between cytokine profiles and malaria susceptibility in primigravid women in the early postpartum period in a high-transmission setting. We hypothesize that alterations in cytokine balance may predispose women to increased vulnerability to malaria after childbirth.

## Methods

### Study site

This study was conducted in the Nanoro Health and Demographic Surveillance System (HDSS) coverage area, located approximately 85 km northwest of Ouagadougou, the capital of Burkina Faso. Malaria transmission in this region is highly seasonal, with intense transmission during the rainy season, from June to October and a peak towards the end of the rainy season.

## Ethical approval

The institutional review board of Centre Muraz/IRSS granted ethical approval for this study (Reference A007-2014/CEI-CM dated 12[th] February 2014). The approval was amended on July 25, 2024, and subsequently extended on September 29, 2015, for an additional year of follow-up for women (July 25, 2014 – July 25, 2016).

## Study design and sample collection

This study aimed to identify cytokine types associated with malaria infection during the postpartum period. It was nested within a larger clinical trial, the COSMIC study, which investigated the efficacy of different preventive strategies against malaria in pregnancy (trial registration numbers: ISRCTN372259296 and NCT01941564) [19]. Participants in the current study were recruited from 02/05/2015 to 30/04/2016 to cover both low and high malaria transmission seasons. Pregnant women were identified from the control group of the COSMIC study, which followed the national policy of administering intermittent preventive treatment in pregnancy with sulfadoxine-pyrimethamine (IPTp-SP). Written informed consent was obtained from all participants, and they were followed up for 3 months after delivery. The characteristics of study participants are described elsewhere [20]. Data on pregnancy background, including the number of IPTp-SP doses and history of malaria episodes, were obtained from the main study.

Ten milliliters of intravenous peripheral blood was collected in heparin tubes from each participant at delivery, and 1 and 3 months post-delivery. Blood samples were processed within 4 hours of collection, and sera were stored at −80°C until analysis. To prepare dried blood spots (DBS) for polymerase chain reaction (PCR) analysis, two 50-μl drops of blood were placed on filter paper (Whatman®) and allowed to dry completely for at least 4 hours at room temperature (25°C). To prepare plasma samples used for cytokine analysis, we carefully placed the heparinized tubes in a swing bucket centrifuge and spin them at 1800 rpm using a Hettich ROTANTA 460R centrifuge for 10 min at room temperature.

At each visit, women were systematically screened for malaria infection using the PfHRP2 rapid diagnostic test (RDT; SD-Bioline) following the manufacturer's instructions for case management. The diagnostic test result was confirmed by light microscopy (LM) and PCR.

Maternal peripheral blood was also collected at delivery to measure hemoglobin (Hb) level using a Hemocue (Hb 301, Sweden); anemia was defined as Hb ≤ 11g/dl.

## *Plasmodium falciparum* infection diagnosis by microscopy

Blood slides were prepared from samples collected at delivery, and at one and three months post-delivery. They were stained with Giemsa 3% for 45–60 min and examined by two independent microscopists blinded to RDT results. Parasite density was calculated against 200 leukocytes, or 500 leukocytes if less than 10 parasites were counted per 200 leukocytes. Slides were considered negative if no parasite was seen after examining 100 high-power fields. In case of discrepancies, a third independent reader's opinion was required, and the final result was based on the two closest readings.

## Malaria diagnosis by nested-PCR

DNA was extracted from DBS using the QI Aamp DNA Extraction Mini-kit (QIAGEN, Valencia, CA, USA) and stored at −20°C until PCR amplification. A nested PCR for amplification of *Plasmodium falciparum* msp2 was performed in 25 μl reaction volumes, using 5 μl of DNA extract for the first round and 1 μl of first-round product for the second round with family-specific primers [21,22]. PCR products were visualized by ethidium bromide-stained agarose gel electrophoresis and UV transillumination, with fragment sizes estimated using Photo CaptMW (v11.01). End-point PCR cycling conditions included primary denaturation at 94°C for 5 minutes, followed by 36 cycles of denaturation at 94°C for 1 min, annealing at 58°C for 2 min, and extension at 72°C for 2 minutes, with a final extension at 72° for 10 minutes. This nested-PCR targeting MSP2 was optimized for detection of *P. falciparum* infection but was not designed for genotyping or clone

differentiation and thus cannot determine whether infections were newly acquired or persistent across delivery and postpartum.

## Cytokine ELISA

Commercially available cytokine ELISA kits for human samples were used to determine the levels of IL-4 (eBioscience BMS225/2), IL-10 (Invitrogen KHC0101), TNF-α (Invitrogen KHC3011), IL-6 (BioSource Europe KAC1261), and INF-γ (Invitrogen KAC1231). Each plate included a recombinant human cytokine standard curve and known positive and negative controls. All specimens were measured in duplicate, and the mean of the two values was taken. The lower limit of detection for each cytokine assay was 15 pg/mL.

## Placental malaria

Placental biopsy samples (2 cm × 2 cm × 1 cm) were collected from the maternal side at delivery, fixed in 10% neutral buffered formalin, and embedded in paraffin wax for histological analysis. Slides were stained with hematoxylin-eosin and read by trained microscopists. Placental infection was classified as acute (parasites present, malaria pigment absent), chronic (parasites and pigment present), past infection (no parasites, no pigment present), or no infection (no parasites or pigment) [23].

## Assessment of dystocia during labor

Dystocia during labor was diagnosed using various criteria in a partograph. A prolonged second stage of labor was documented, lasting more than 2 hours for nulliparous women [24], 3 hours with an epidural [25], and 1 hour for multiparous women. The reasons for prolonged labor were classified as mechanical or dynamic and could include weak contractions, fetal malposition, fetal macrosomia [24] cephalopelvic disproportion, and other medical or non-medical conditions. Other factors including maternal exhaustion, and pre-existing medical conditions were investigated. Blood loss during and immediately after delivery was estimated and documented, with categories of normal (< 500 mL for vaginal deliveries and < 1000 mL for cesarean sections) and excessive (indicating hemorrhage).

## Sample size and statistical analysis

The study included 33 primigravid women enrolled at delivery and followed up one and three months after delivery. The sample size (33 primigravida women) was determined based on the feasibility of the COSMIC study and the logistical constraints of participant follow-up. Although this sample size may be considered modest, it is consistent with previous immunological studies examining cytokine responses in malaria-infected pregnant women [26]. Given the observed differences in IL-10 levels between infected and non-infected women (σ = 0.9, p < 0.001), post-hoc power analysis indicates that our study had > 80% power to detect significant differences in cytokine expression at an alpha level of 0.05. Future studies with larger cohorts will be needed to confirm these results in broader populations.

Data were analyzed using Stata Statistical Software: version 15 (College Station, Texas). The Shapiro-Wilks W test was used to assess the normality of continuous variables, with logarithmic transformations applied where necessary. Spearman's rho test was used to determine associations between continuous variables. Baseline cytokine levels in infected and uninfected women were compared using the Mann-Whitney test, unpaired Student's t-test, or Kruskal-Wallis test.

To complement p-values, effect sizes were calculated using Cohen's d to assess the magnitude of differences between cytokine levels in malaria-infected and non-infected women. The following formula was used: $d = ((M1 - M2)/SDpooled)$, where $SDpooled = \sqrt{(((S[D1]^2 + S[D2]^2))/2)}$.

Effect sizes were classified as small (d = 0.2), medium (d = 0.5), and large (d ≥ 0.8). This analysis provided additional insights into the clinical relevance of cytokine differences beyond statistical significance.

Plasma cytokine concentrations were measured by enzyme-linked immunosorbent assay (ELISA) and analyzed using the "Cluster analysis" method [27], which requires a minimum of 5–10 individuals per group. Malaria infection status was determined by polymerase chain reaction (PCR) and microscopy. This dataset allowed a comprehensive examination of the relationships between cytokine profiles and malaria susceptibility.

## Results

### Characteristics of the study population

A total of 33 primigravid women were enrolled in this study, with a median age of 19 years (range, 18–20.5 years). At delivery, *P. falciparum* malaria infection was confirmed by PCR in 48.5% (n = 16) of women, while peripheral infection was determined by microscopy in 9.1% (n = 3). The mean parasite density at delivery was 540.9 [95% CI: 1–5,584] parasites/µL. Anemia was prevalent, with 60.6% (n = 20) of women anemic at delivery, but this rate decreased to 15.2% (n = 5) at one month postpartum before rising to 24.2% (n = 8) at three months postpartum. Placental malaria was detected in 23.8% (n = 5) of women at delivery (Table 1).

### Cytokine distributions and correlations at delivery

At delivery, the levels of pro-inflammatory cytokines (IL-6, TNF-α) and anti-inflammatory cytokines (IL-4, IL-10) were analyzed to determine their correlations. Spearman's rho test showed significant positive correlations between most cytokines. The strongest correlation was observed between IL-6 and IL-10 ($\sigma = 0.8$, $p < 0.001$) (Fig 1), followed by IL-10 and IL-4 ($\sigma = 0.7$, $p < 0.001$), and IL-6 and IL-4 ($\sigma = 0.6$, $p < 0.001$). Moderate correlations were found between IL-6 and TNF-α

Table 1. Characteristics of the study population characteristics.

| Characteristics | Values |
|---|---|
| **Age—years, Median** (IQR) | 19 (18 - 20.5) |
| **Anaemia** | |
| Delivery (Anemic) —n (%) | 20 (60.6) |
| 1 month after delivery (Anemic) —n (%) | 5 (15.2) |
| 3 months after delivery (Anemic) —n (%) | 8 (24.2) |
| **Parasite density of peripheral infection (N = 17)\* —** Mean (min – max) Parasite/µL | |
| Delivery **(Infected) —n = 03** | 540.9 (0 - 5584.0) |
| **Malaria infection\*\* —n (%)** | |
| Delivery (Infected) | 16 (48.5) |
| 1 month after Delivery (Infected) | 3 (9.7) |
| 3 months after Delivery (Infected) | 7 (21.9) |
| **Placental malaria\*\*\* at delivery (N = 21) —n (%)** | |
| Active | 1 (4.8) |
| Past only | 4 (19.1) |
| No infection | 16 (76.2) |
| **Conditions of delivery (**Dystocia**) —n (%)** | 6 (18.2) |
| **Period of delivery (**transmission season**) — n (%)** | |
| High | 20 (60.6) |
| Low | 13 (39.4) |

\*determined by microscopy among the 17 malaria – infected women during pregnancy; \*\*as determine by PCR; \*\*\* determined by placental histology using microscopy

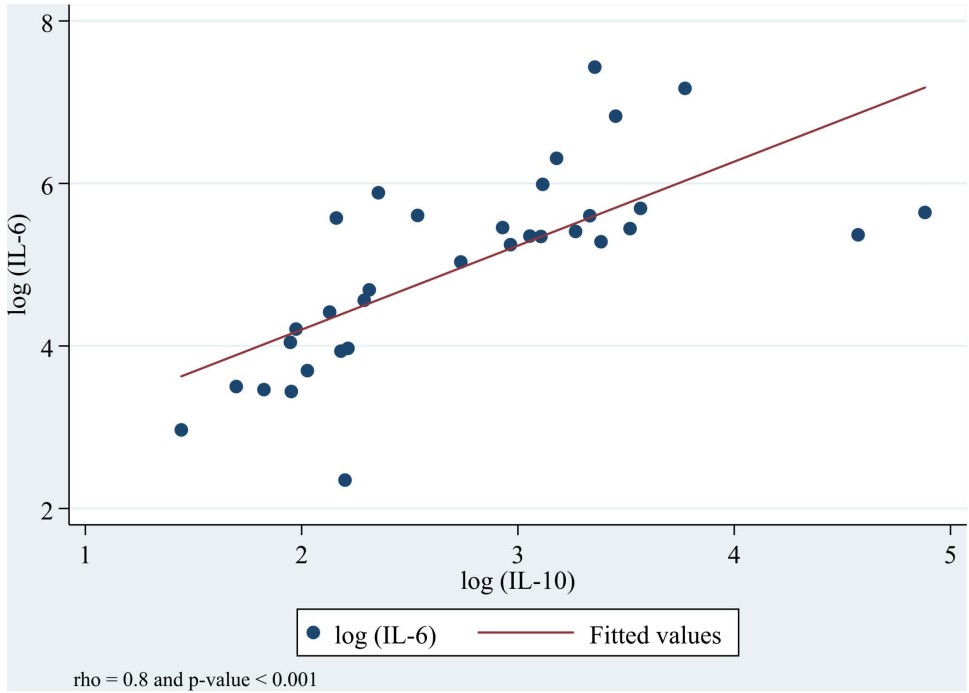

**Fig 1. Relationship between plasma IL-6 and IL-10 concentrations at delivery.** The strongest correlation result following the Spearman's correlation test coefficients is presented in this figure. Spearman's correlation graded interpretation is defined as follows: (i) rho between 0.1 - 0.3 = weak; (ii) rho between 0.4 - 0.7 = moderate and (iii) rho between 0.8 - 1.0 = strong correlation.

($\sigma = 0.5$, $p = 0.003$), TNF-α and IL-4 ($\sigma = 0.5$, $p = 0.002$), and between IFN-γ and IL-4 ($\sigma = 0.5$, $p = 0.007$). In addition, IL-10 showed a moderate correlation with TNF-α ($\sigma = 0.4$, $p = 0.013$) and IFN-γ ($\sigma = 0.4$, $p = 0.0169$), and a weak correlation was observed between IL-6 and IFN-γ ($\sigma = 0.3$, $p = 0.075$).

To complement p-values, Cohen's d was calculated to assess the effect sizes of cytokine differences between malaria-infected and non-infected women. IL-4 showed a medium-to-large effect size ($d = 0.78$), suggesting a notable difference between groups. IL-10 and IL-6 had medium effect sizes ($d = 0.65$ and $d = 0.52$, respectively), supporting their role in immune modulation during infection. In contrast, TNF-α ($d = 0.08$) and IFN-γ ($d = 0.06$) had small effects, indicating limited differences between infected and non-infected women (Table 2).

## Comparison of cytokine levels between infected and non-infected women at delivery

At delivery, plasma levels of pro-inflammatory cytokines (IL-6: $p < 0.001$; TNF-α: $p = 0.005$; IFN-γ: $p = 0.003$) and anti-inflammatory cytokines (IL-4: $p < 0.001$; IL-10: $p < 0.001$) were significantly higher in malaria-infected women compared to non-infected women within the same cohort. Conversely, the IL-6: IL-10 ratio was significantly higher in non-infected women ($p = 0.005$) (Fig 2a–2e).

## Pro- and anti-inflammatory cytokine balances during the postpartum period

The IL-6: IL-10 ratio was significantly higher in non-infected women at delivery [1.98 (1.81–2.15)] than in infected women [1.72 (1.57–1.87)], $p = 0.019$. However, this ratio did not differ significantly between infected and non-infected women, as was the PCR diagnosis at one and three months postpartum (Table 3).

Table 2. Correlations between cytokines concentrations (Spearman's rho correlation coefficients), pooled SD, and effect sizes.

| Cytokine | IL-10 | IL-6 | TNF-α | IL-4 | IFN-γ | Pooled SD | Cohen's d | Effect size interpretation |
|---|---|---|---|---|---|---|---|---|
| **IL-10** | 1 | 0.8[a] (p<0.001)[b] | 0.4 (p=0.013) | 0.7 (p<0.001) | 0.4 (p=0.0169) | 21.96 | 0.65 | Medium |
| **IL-6** | 0.8 (p<0.001) | 1 | 0.5 (p=0.003) | 0.6 (p<0.001) | 0.3 (p=0.075) | 335.19 | 0.52 | Medium |
| **TNF-α** | 0.4 (p=0.013) | 0.5 (p=0.003) | 1 | 0.5 (p=0.002) | 0.5 (p=0.007) | 274.71 | 0.08 | Small |
| **IL-4** | 0.7 (p<0.001) | 0.6 (p<0.001) | 0.5 (p=0.002) | 1 | 0.5 (p=0.007) | 16.52 | 0.78 | Medium-Large |
| **IFN-γ** | 0.4 (p=0.0169) | 0.3 (p=0.075) | 0.5 (p=0.007) | 0.5 (p=0.007) | 1 | 0.0289 | 0.06 | Small |

**a**: Spearman's rho (correlation coefficient); **b**: p-value.

## Comparison of cytokine levels during postpartum period

One month after delivery, infected women had significantly higher IL-10 levels (p=0.012). Overall, cytokine levels were generally higher in infected women, although not all differences were statistically significant. At three months postpartum, cytokine levels were slightly higher in non-infected women except for IL-4, which remained higher in infected women, but without significant differences (Table 4).

## Relationships between cytokine concentrations and baseline characteristics

Cytokine levels at delivery were strongly correlated with malaria infection status, with IL-10 (σ=0.9, p<0.001) and IL-4 (σ=0.9, p<0.001) showing the strongest correlations. Parasite count also moderately correlated with IL-10 (σ=0.5, p=0.008). A negative correlation was observed between malaria infection status and the IL-6: IL-10 ratio (σ=−0.5, p=0.004), and between parasite count and the IL-6: IL-10 ratio (σ=−0.4, p=0.012) (Table 5).

## Discussion

This study showed that malaria-infected women had significantly higher levels of both pro-inflammatory (IL-6, TNF-α, IFN-γ) and anti-inflammatory (IL-4, IL-10) cytokines at delivery compared to non-infected women. The IL-6: IL-10 ratio was notably lower in infected women, suggesting a dominance of anti-inflammatory responses during malaria infection. These cytokine balances shifted during the postpartum period, with non-infected women showing higher IL-6: IL-10 ratios at 1 and 3 months postpartum, reflecting immune rebalancing after delivery.

The elevated levels of IL-10 observed in malaria-infected women at delivery are consistent with a substantial body of literature demonstrating increased IL-10 production during *P. falciparum* infection in pregnancy, in both peripheral and placental compartments [28–31]. These studies collectively support the role of cytokines as a key immunoregulatory cytokine that limits excessive inflammation while shaping susceptibility to persistent infection.

Previous work has shown that IL-10 concentrations are positively associated with parasite density and placental infection, and are often more pronounced in primigravid women, who lack parity-associated immunity [31–33]. Our findings extend these observations by demonstrating that this IL-10 dominated immune profile persists into the postpartum period among infected women [26,34].

While IL-10 may protect against immunopathology, excessive or sustained IL-10 production can suppress effective Th1-mediated immune responses required for parasite clearance [28,35]. This immunoregulatory bias may contribute to parasite persistence and delayed clearance around delivery, potentially explaining the sustained cytokine alterations observed postpartum.

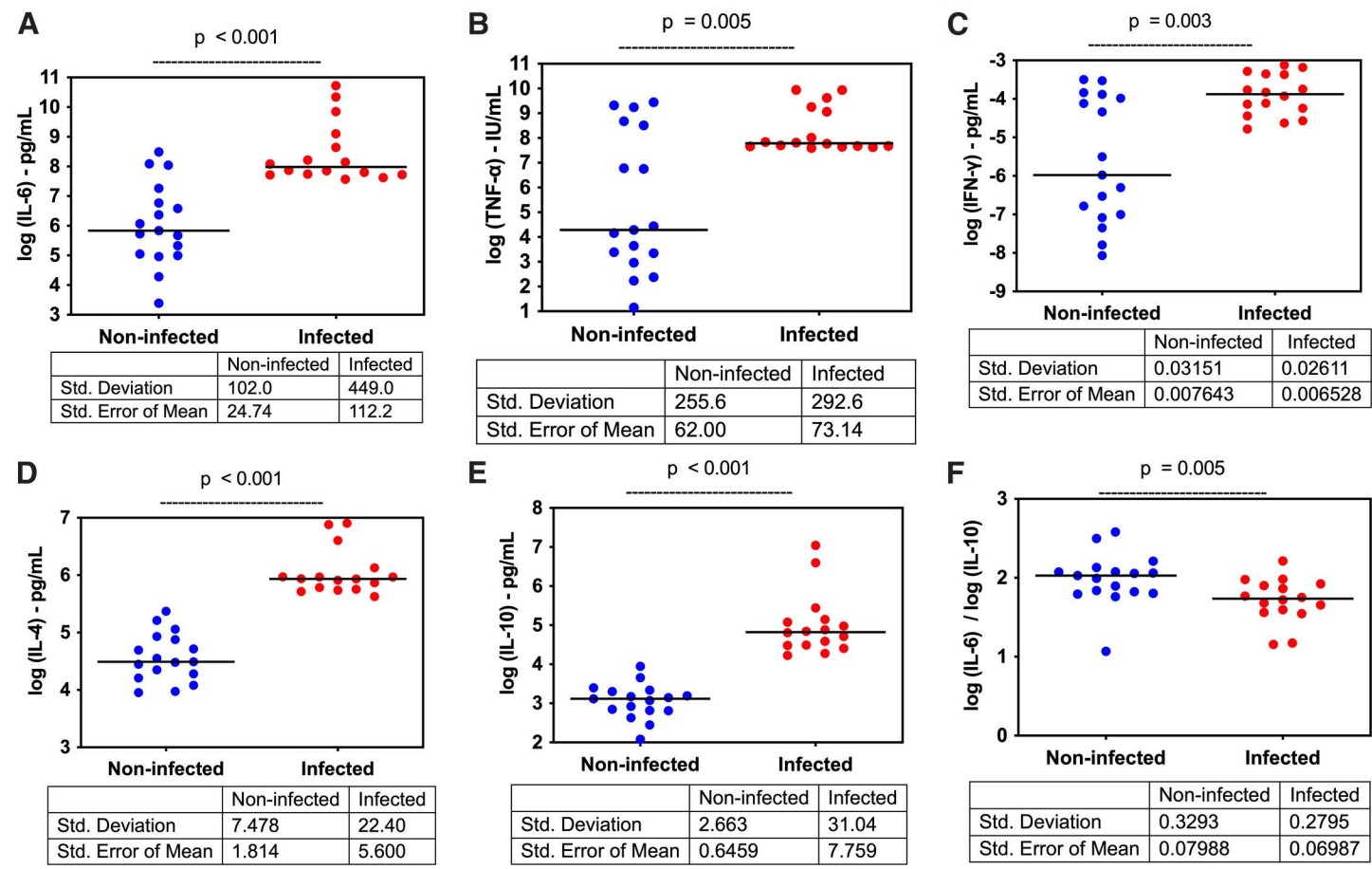

**Fig 2. Comparison of cytokine levels between infected women and non-infected women at delivery.** This figure illustrates the differences in cytokine levels between infected and non-infected women at delivery, highlighting key cytokines involved in the immune response to malaria. (2a) IL-6: Infected women show significantly higher IL-6 levels compared to non-infected women (p-value < 0.001), indicating a heightened pro-inflammatory response. (2b) TNF-α: TNF-α levels are also elevated in infected women (p-value = 0.005), further reflecting an increased inflammatory response. (2c) IFN-γ: There is a significant increase in IFN-γ levels in infected women (p-value = 0.003), suggesting enhanced activation of cellular immunity. (2d) IL-4: Infected women exhibit higher IL-4 levels (p-value < 0.001), indicating a concurrent anti-inflammatory response. (2e) IL-10: The levels of IL-10, a key anti-inflammatory cytokine, are significantly elevated in infected women (p-value < 0.001), which may contribute to immune modulation during malaria infection. (2f) IL-6/ IL-10 Ratio: The IL-6/IL-10 ratio is significantly lower in infected women (p-value = 0.005), indicating a relative dominance of anti-inflammatory over pro-inflammatory responses in these individuals.

**Table 3. Pro- and anti-inflammatory cytokines balances at delivery, 1 month and 3 months after delivery.**

|  | IL-6 to IL-10 ratios | | |
|---|---|---|---|
|  | Non-infected | Infected | p |
| **Delivery** | n = 17 | n = 16 |  |
|  | 1.98 (1.81 - 2.15) | 1.72 (1.57 - 1.87) | **0.019** |
| **1 Month after delivery** | n = 28 | n = 3 |  |
|  | 1.81 (1.54 - 2.09) | 1.56 (0.72 - 2.41) | 0.558 |
| **3 Months after delivery** | n = 25 | n = 7 |  |
|  | 1.88 (1.65 - 2.11) | 1.49 (1.01 - 1.97) | 0.105 |

p-values were calculated using t-test on log-transformed variables.

**Table 4. Cytokines profiles (geometric mean (95% CI)) at 1 month and 3 months after delivery.**

| Cytokines | Primiparous at 1 month after delivery | | | Primiparous at 3 months after delivery | | |
|---|---|---|---|---|---|---|
| | Non-infected (n = 28) | Infected (n = 3) | p | Non-infected (n = 25) | Infected (n = 7) | p |
| IL-6 (pg/mL) | 66.57 (34.21 - 129.57) | 238.43 (113.27 - 501.89) | 0.150 | 260.73 (103.75 - 655.23) | 74.53 (12.35 - 449.87) | 0.327 |
| TNF-α (pg/mL) | 54.01 (25.39 - 114.90) | 198.75 (97.43 - 405.44) | 0.242 | 104.81 (44.93 - 244.49) | 41.33 (9.60 - 177.83) | 0.494 |
| IFN-γ (IU/mL) | 0.05 (0.03 - 0.09) | 0.09 (0.06 - 0.17) | 0.664 | 0.10 (0.06 - 0.17) | 0.06 (0.02 - 0.16) | 0.121 |
| IL-4 (pg/mL) | 48.99 (37.47 - 64.08) | 77.35 (61.45 - 97.38) | 0.170 | 35.57 (26.35 - 48.00) | 31.85 (18.34 - 55.33) | 0.820 |
| IL-10 (pg/mL) | 10.56 (8.44 - 13.22) | 36.39 (7.89 - 167.72) | **0.012** | 19.56 (13.18 - 29.02) | 19.53 (6.40 - 59.58) | 0.715 |

\* p-values were calculated using t-test on log transformed values of parameters

**Table 5. Relationships between plasma cytokine concentrations and delivery.**

| Cytokines | Women status at delivery | | | | | | | |
|---|---|---|---|---|---|---|---|---|
| | Malaria | | Dystocia | | Anaemia | | Parasitaemia | |
| | σ | p | σ | p | σ | p | σ | p |
| IL-6 | 0.7 | **<0.001** | 0.2 | 0.213 | −0.2 | 0.293 | 0.3 | 0.148 |
| IL-10 | 0.9 | **<0.001** | 0.1 | 0.493 | 0.0 | 0.971 | 0.5 | **0.008** |
| IL-4 | 0.9 | **<0.001** | 0.1 | 0.409 | 0.0 | 0.857 | 0.2 | 0.288 |
| IFN-γ | 0.5 | **0.002** | 0.2 | 0.359 | −0.1 | 0.614 | 0.3 | 0.135 |
| TNF-α | 0.5 | **0.004** | 0.0 | 0.784 | −0.1 | 0.516 | 0.1 | 0.457 |
| IL-6: IL-10 | −0.5 | **0.004** | 0.0 | 0.820 | 0.0 | 0.943 | −0.4 | **0.012** |

The lower IL-6: IL-10 ratio observed in malaria-infected women at delivery indicates an anti-inflammatory bias that may facilitate immune evasion by *P. falciparum* [34,36]. IL-10 has been shown to inhibit macrophage activation, antigen presentation, and pro-inflammatory cytokine production, thereby limiting effective parasite elimination [29,30,37]. The schematic presented in Fig 3 illustrates this proposed cytokine balance shift, highlighting how IL-10 mediated regulation may allow parasite persistence while preventing excessive inflammation.

It is important to note that although we used nested PCR targeting the polymorphic *msp2* gene to increase diagnostic sensitivity, this assay was implemented strictly for detection and not for genotyping. As such, we could not distinguish whether postpartum infections represented newly acquired infections or persistent infections originating before delivery. Previous studies have demonstrated that *msp2* genotyping is a powerful tool to differentiate persistent from new infections in longitudinal malaria studies [38,39]. Incorporation of such genotyping approaches in future studies would help clarify whether sustained IL-10 production postpartum is specifically linked to persistent parasitemia.

The postpartum period represents a critical window for immune reconstitution as the maternal immune system transitions from pregnancy-induced tolerance to a more balanced state [10]. The higher IL-6: IL-10 ratios in non-infected women during this period suggest a recovery of pro-inflammatory responses, essential for combating infections [40]. This shift in cytokine balance towards a pro-inflammatory state might explain the reduced susceptibility to malaria observed in non-infected women as they recover from pregnancy. Furthermore, regulatory T cells (Tregs) may also contribute to this

 

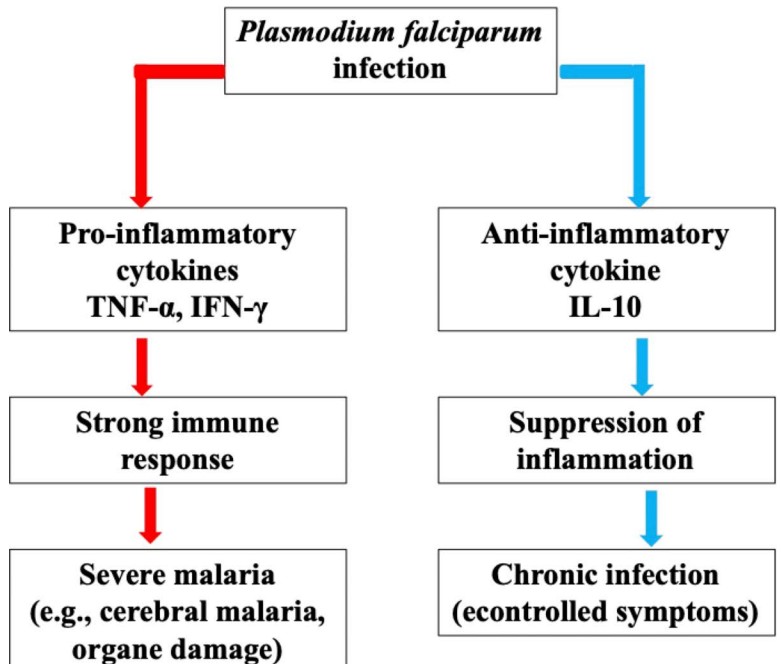

**Fig 3. Cytokine balance in *Plasmodium falciparum* infection.** The immune response to *Plasmodium falciparum* infection involves a delicate balance between pro-inflammatory and anti-inflammatory cytokines. Pro-inflammatory cytokines such as TNF-α and IFN-γ trigger a strong immune response, which is essential for parasite clearance but also increases the risk of severe inflammation and tissue damage. In contrast, IL-10 suppresses excessive inflammation, reducing immune-mediated pathology but facilitating parasite persistence. The clinical outcome depends on this cytokine balance: a dominant pro-inflammatory response can lead to severe malaria (e.g., cerebral malaria, organ damage), whereas a high IL-10 response may promote chronic infection with controlled symptoms.

immunological phenotype. IL-10–producing Tregs have been implicated in dampening antigen-specific immune responses and promoting chronic parasitemia during malaria infection [41]. The spread of these cells in postpartum malaria-infected women might help to explain extended susceptibility.

Physical factors such as dystocia were also considered in this study (as shown in Fig 4a–4e). Although dystocia was associated with higher cytokine levels in infected women, it did not significantly alter the overall cytokine balance between infected and non-infected groups. This suggests that while physical stressors contribute to inflammation, malaria infection itself is the primary driver of cytokine changes [42,43].

The use of the cluster analysis approach ensured sufficient statistical power to detect significant differences in cytokine levels between infected and non-infected women [27].

Our findings highlight the complex relationship between inflammatory cytokines and malaria infection status during postpartum, suggesting that cytokine profiles may serve as biomarkers for malaria pathogenesis in this population. Therapeutic approaches targeting cytokine modulation could offer novel strategies to improve clinical outcomes. IL-4, for instance, has been shown to reduce parasitaemia, cerebral malaria pathology, and mortality by promoting parasite clearance while reducing brain inflammation [44]. Additionally, rosiglitazone, a PPARγ agonist, enhances phagocytic clearance of parasitized erythrocytes and reduces inflammatory responses in experimental cerebral malaria by inhibiting specific signaling pathways [45].

Furthermore, elevated IFN-γ and TNF-α levels have been associated lower fever and malaria risk, while IL-12 production plays a role in reducing parasitemia. Conversely, an imbalanced proinflammatory-to-anti-inflammatory cytokine ratio

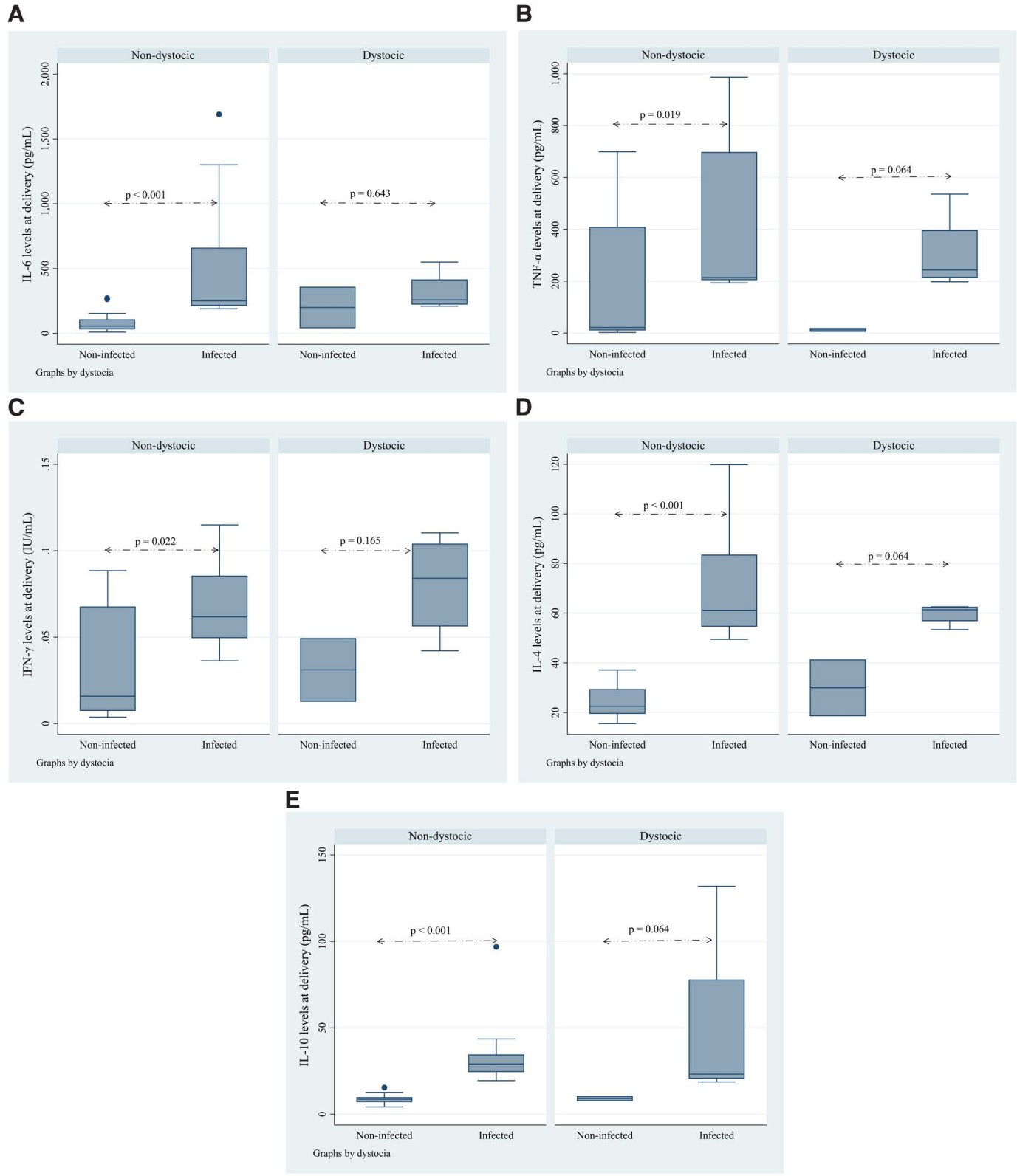

**Fig 4. Assessment of the effect of dystocia on cytokine levels at delivery.** This illustration reports the effect of dystocia on cytokine levels at delivery in women infected and non-infected by *P. falciparum*. IL-6 levels (4a) are significantly higher in infected women with dystocia compared to

non-infected women (p < 0.001). TNF-α (4b) also shows a significant increase in the infected group (p < 0.05). IFN-γ levels (4c) are elevated in infected women, but the difference is not statistically significant. Both IL-4 (4d) and IL-10 (4e) levels are significantly higher in infected women, with IL-10 showing the most substantial difference (p < 0.001). This suggests that dystocia impacts cytokine levels, particularly in infected women, indicating an enhanced inflammatory response.

may increase the risk of severe malaria symptoms [46]. These findings underscore the potential of targeting cytokine profiles as a therapeutic avenue.

Our study highlights key insights, but further research is needed to identify the cellular sources of IL-10 and its role in postpartum malaria immunity. Cytokine profiling, particularly IL-10-centered ratios, may serve as valuable biomarkers for malaria pathogenesis and immune recovery in the postpartum period. Future research should focus on (i) PBMC stimulation assays to verify whether IL-10 suppresses protective inflammatory responses.; (ii) Treg profiling by flow cytometry to examine the role of regulatory T cells in malaria persistence; (iii) RNA-seq or proteomic analyses to study signaling pathways downstream of cytokine modulation; and (iv) Clinical trials with IL-10 inhibitors to test their effectiveness in antimalarial immunity without increasing inflammation.

Furthermore, studying the impact of interventions aimed at modulating cytokine responses could offer promising avenues for improving maternal and fetal health in malaria-endemic regions.

## Conclusion

This study shows that women infected with malaria-postpartum exhibit a distinct cytokine profile, with elevated IL-10 and a suppressed IL-6:IL-10 ratio at delivery. These findings suggest an anti-inflammatory bias that may impair immune clearance and promote parasite persistence. In contrast, non-infected women regained their pro-inflammatory responses after delivery. The observed cytokine shifts highlight the potential of immune profiling as a biomarker for malaria risk and support further research into immunomodulatory therapies for postpartum malaria.

## Supporting information

**S1 Fig. Traore et al_ graphical abstract _Plos One.**
(DOCX)

**S1 File. Traore et al_minimal data set – updated.**
(XLSX)

## Acknowledgments

We are grateful to the COSMIC trial participants and the field staff of the Clinical Research Unit of Nanoro for their contribution to the study completion. We would like to thank the nulliparous women recruited in the HDSS catchment area.

## Author contributions

**Conceptualization:** Ousmane Traore.

**Formal analysis:** Toussaint Rouamba.

**Funding acquisition:** Henk D. F. H. Schallig, Halidou Tinto.

**Methodology:** Ousmane Traore, Serge Henri Zango.

**Supervision:** Maminata Traore-Coulibaly, Henk D. F. H. Schallig, Halidou Tinto.

**Validation:** Ousmane Traore, Innocent Valea.

**Writing – original draft:** Ousmane Traore.

**Writing – review & editing:** Toussaint Rouamba, Serge Henri Zango, Hermann Sorgho, Innocent Valea, Maminata Traore-Coulibaly, Henk D. F. H. Schallig, Halidou Tinto.

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
