## [Decision Letter · Decision Letter 0]

19 Nov 2025

Dear Dr. TRAORE,

Thank you for submitting your manuscript to PLOS ONE. After careful consideration, we feel that it has merit but does not fully meet PLOS ONE’s publication criteria as it currently stands. Therefore, we invite you to submit a revised version of the manuscript that addresses the points raised during the review process.

We look forward to receiving your revised manuscript.

Kind regards,

David J. Diemert, M.D.

Academic Editor

PLOS ONE

Journal Requirements:

 “This study received financial support from the Belgium cooperation (DGD-ITM Framework Agreement 4 – FA4 2017-2021), and from the European Union (FP7-Health-F3-305662).”        

Reviewers' comments:

Reviewer's Responses to Questions

**Comments to the Author**

1. Is the manuscript technically sound, and do the data support the conclusions?

Reviewer #1: Partly

2. Has the statistical analysis been performed appropriately and rigorously?

Reviewer #1: Yes

3. Have the authors made all data underlying the findings in their manuscript fully available?

Reviewer #1: Yes

4. Is the manuscript presented in an intelligible fashion and written in standard English?

Reviewer #1: Yes

Reviewer #1: Major points

There is an abundance of published information concerning the associations between infection during pregnancy with P. falciparum and elevated plasma cytokine levels, both in peripheral and placental compartments. Much of the relevant literature is not cited by the authors. This should be rectified. Particularly pertinent to the authors' findings are those studies that have clearly identified elevated levels of IL-10 in infected mothers.

Otherwise, there is one essential piece of information that the authors have not included in their manuscript, for reasons that are not at all clear. They identified infections in peripheral blood through the use of different techniques, including, most importantly, a PCR-based method. The particular method employed, relying on amplification of polymorphic regions of the MSP2 gene, would, I assume, allow for confirmation of whether or not infections detected in the post-partum period comprised newly-acquired infections or - perhaps most likely - infections that were already present prior to delivery that had persisted post-delivery. In the latter case, I would conjecture that it would be no surprise at all to find that the (anti-inflammatory) IL-10-dominated cytokine profile observed prior to delivery in infected mothers was, as it were, carried over - persisted, along with the infection - into the post-partum period in those that failed to clear their infections at or around the time of delivery. The authors should therefore consider including the relevant PCR-derived information in the Results, with an appropriate modification of the Discussion.

Minor points to address

1. Table 1: Line 180 of Results states that the the mean parasite density at delivery was 540.9, but a different number appears in Table 1 - this anomaly should be corrected, and the numbers of individuals with peripheral infections identified by microscopy should also be added to the Table; an * is missing for Parasite density of peripheral infection at delivery; a third * is missing for Placental malaria at delivery

2. In various places throughout the manuscript (Methods, Table 5, Figure 1), the authors erroneously refer to the use of serum samples, whereas they were actually using plasma. This should be corrected throughout

**Do you want your identity to be public for this peer review?** For information about this choice, including consent withdrawal, please see our Privacy Policy

Reviewer #1: **Yes: ** Adrian J F Luty

---

## [Author Response · Author response to Decision Letter 1]

4 Jan 2026

Response to reviewers

Manuscript title: Postpartum cytokine shifts and IL-10–mediated immune suppression in malaria-infected primigravid women

Manuscript number: PONE-D-25-29041

Dear Academic Editor and Reviewer,

We thank you sincerely for your thoughtful and constructive review of our manuscript. We have carefully addressed each point raised, and we appreciate the opportunity to improve the clarity, rigor, and contextual relevance of our work. Below is a point-by-point response to all reviewer comments and editorial requests.

Reviewer #1 Comments

Major Comments

Comment 1:

There is an abundance of published information concerning the associations between infection during pregnancy with P. falciparum and elevated plasma cytokine levels, both in peripheral and placental compartments. Much of the relevant literature is not cited by the authors. This should be rectified. Particularly pertinent to the authors' findings are those studies that have clearly identified elevated levels of IL-10 in infected mothers.

Otherwise, there is one essential piece of information that the authors have not included in their manuscript, for reasons that are not at all clear.

Author response:

We thank the reviewer for this important observation and agree that our findings should be more clearly contextualized within the existing literature.

We have revised the Discussion section to include and synthesize key studies documenting elevated IL-10 levels and broader cytokine alterations in Plasmodium falciparum - infected pregnant women, in both peripheral and placental compartments.

These additions better situate our results within established evidence on the immunoregulatory role of IL-10 during pregnancy-associated malaria and strengthen the interpretation of our findings.

Comment 2:

They identified infections in peripheral blood through the use of different techniques, including, most importantly, a PCR-based method. The particular method employed, relying on amplification of polymorphic regions of the MSP2 gene, would, I assume, allow for confirmation of whether or not infections detected in the post-partum period comprised newly-acquired infections or - perhaps most likely - infections that were already present prior to delivery that had persisted post-delivery. In the latter case, I would conjecture that it would be no surprise at all to find that the (anti-inflammatory) IL-10-dominated cytokine profile observed prior to delivery in infected mothers was, as it were, carried over - persisted, along with the infection - into the post-partum period in those that failed to clear their infections at or around the time of delivery. The authors should therefore consider including the relevant PCR-derived information in the Results, with an appropriate modification of the Discussion.

Author response:

We sincerely thank the reviewer for this highly insightful comment. We agree that distinguishing whether postpartum infections were newly acquired or persistent from the antenatal period is essential for interpreting the observed cytokine dynamics, especially the sustained IL-10 production.

In response, we have added clarifications in the Methods and Discussion sections to emphasize that while our nested PCR assay targeted polymorphic regions of the msp2 gene, it was optimized solely for sensitive detection of Plasmodium falciparum, not for genotyping. The protocol was not designed to differentiate parasite clones or determine infection multiplicity and persistence across time points.

We now explicitly acknowledge this limitation in the manuscript and highlight the utility of msp2 genotyping to distinguish between new and persistent infections in longitudinal malaria studies. Incorporating such genotyping in future studies would allow a more definitive assessment of infection dynamics and their immunological correlates, including whether sustained IL-10 expression postpartum is linked to persistent parasitaemia. This would directly test the reviewer’s well-argued hypothesis regarding IL-10 carry-over from unresolved infections at delivery.

These points have been added to the Discussion to strengthen the interpretation of our findings in light of this important methodological consideration.

Minor Comments

Comment 3:

Table 1: Line 180 of Results states that the mean parasite density at delivery was 540.9, but a different number appears in Table 1. This anomaly should be corrected, and the numbers of individuals with peripheral infections identified by microscopy should also be added to the Table. An * is missing for “Parasite density of peripheral infection at delivery”; a third * is missing for “Placental malaria at delivery”.

Author response:

We thank the reviewer for pointing out this inconsistency and the missing annotations. We have addressed all points as follows:

• Mean parasite density at delivery:

We corrected the discrepancy by updating the value in the Results section to match the value reported in Table 1, based on the microscopy-confirmed data.

• Microscopy-positive infections:

We added the number of microscopy-detected peripheral infections at delivery to Table 1, as requested.

• Asterisks for clarification:

We have now included:

o A single asterisk (*) next to the line reporting parasite density of peripheral infection, indicating that it was determined by microscopy.

o A third asterisk (***) for placental malaria, to match the corresponding footnote referring to diagnosis by placental histology.

All changes have been incorporated into the revised version of Table 1 and reflected in the updated manuscript.

Comment 4:

In various places throughout the manuscript (Methods, Table 5, Figure 1), the authors erroneously refer to “serum” whereas they were actually using “plasma”. This should be corrected throughout.

Author response:

Thank you for pointing this out. All references to "serum" have been corrected to "plasma" throughout the manuscript, including the Methods, Table 5, Figure 1, and relevant figure legends.

PLOS ONE journal requirements

1. Role of funders

Author response:

We have included the following statement in our cover letter and manuscript:

2. Ethics statement

Author response:

We moved the ethics statement to the Methods section under a new subheading “Ethical Approval.” It reads:

The institutional review board of Centre Muraz/IRSS granted ethical approval for this study (Reference A007-2014/CEI-CM dated 12th February 2014). The approval was amended on July 25, 2024, and subsequently extended on September 29, 2015, for an additional year of follow-up for women (July 25, 2014 – July 25, 2016).

It has been removed from all other sections.

3. Reference list

Author response:

The reference list has been carefully reviewed and updated. No retracted papers are cited. We added several foundational references regarding cytokine responses in malaria-infected pregnant women.

We believe these changes substantially improve the manuscript and respectfully submit our revised version for your consideration. Thank you again for your helpful feedback and support.

4. Figures updated to comply with PLOS format

Author Response:

All figures have been revised and uploaded in TIFF format at 300 dpi resolution, in line with PLOS ONE requirements.

---

## [Decision Letter · Decision Letter 1]

28 Jan 2026

Postpartum cytokine shifts and IL-10–mediated immune suppression in malaria-infected primigravid women

PONE-D-25-29041R1

Dear Dr. TRAORE,

We’re pleased to inform you that your manuscript has been judged scientifically suitable for publication and will be formally accepted for publication once it meets all outstanding technical requirements.

Kind regards,

David J. Diemert, M.D.

Academic Editor

PLOS One

Additional Editor Comments (optional):

Reviewers' comments:

Reviewer's Responses to Questions

**Comments to the Author**

Reviewer #1: All comments have been addressed

2. Is the manuscript technically sound, and do the data support the conclusions?

Reviewer #1: (No Response)

3. Has the statistical analysis been performed appropriately and rigorously?

Reviewer #1: (No Response)

4. Have the authors made all data underlying the findings in their manuscript fully available?

Reviewer #1: (No Response)

5. Is the manuscript presented in an intelligible fashion and written in standard English?

Reviewer #1: (No Response)

Reviewer #1: (No Response)

**Do you want your identity to be public for this peer review?** For information about this choice, including consent withdrawal, please see our Privacy Policy

Reviewer #1: **Yes: ** Adrian JF Luty

---

## [Editor Report · Acceptance letter]

PONE-D-25-29041R1

PLOS One

Dear Dr. Traore,

I'm pleased to inform you that your manuscript has been deemed suitable for publication in PLOS One. Congratulations! Your manuscript is now being handed over to our production team.

Kind regards,

on behalf of

Dr. David J. Diemert

Academic Editor

PLOS One